# Assessing the Representation of Arctic Sea Ice and Marginal Ice Zone in Ocean/Sea Ice Reanalyses

Francesco Cocetta[1], Lorenzo Zampieri[1], Julia Selivanova[1], and Doroteaciro Iovino[1]

[1]CMCC Foundation - Euro-Mediterranean Center on Climate Change, Italy

**Correspondence:** Francesco Cocetta (francesco.cocetta@cmcc.it)

**Abstract.** The recent development of data-assimilating reanalyses of the global ocean and sea ice enables a better understanding of the polar region dynamics and provides gridded descriptions of sea ice variables without temporal and spatial gaps. Here, we study the spatiotemporal variability of the Arctic sea ice area and thickness using the Global ocean Reanalysis Ensemble Product (GREP) produced and disseminated by the Copernicus Marine Service (CMS). GREP is compared and validated against the state-of-the-art regional reanalyses PIOMAS and TOPAZ, and observational datasets of sea ice concentration and thickness for the period 1993–2020. Our analysis presents pan-Arctic metrics but also emphasizes the different responses of ice classes, marginal ice zone (MIZ) and pack ice, to climate changes. This aspect is of primary importance since the MIZ accounts for an increasing percentage of the summer sea ice as a consequence of the Arctic warming and sea ice extent retreat, among other processes. Our results show that GREP provides reliable estimates of present-day and recent past Arctic sea ice states and that the seasonal to interannual variability and linear trends in the MIZ area are properly reproduced, with ensemble spread often being as broad as the uncertainty of the observational dataset. The analysis is complemented by an assessment of the average MIZ latitude and its northward migration in recent years, a further indicator of the Arctic sea ice decline. There is substantial agreement between GREP and reference datasets in the summer. Overall, GREP is an adequate tool for gaining an improved understanding of the Arctic sea ice, also in light of the expected warming and the Arctic transition to ice-free summers.

## 1 Introduction

Arctic sea ice has experienced a rapid decline in extent (Shokr and Ye, 2023), substantial thinning (Sumata et al., 2023), and a loss of multiyear sea ice (Babb et al., 2023) in recent decades with subsequent impacts on climate, human activities, and ecosystem in the region (Meredith et al., 2019). According to 21st-century projections, negative sea ice volume (SIV) and extent (SIE) trends are expected to continue unless anthropogenic greenhouse gas emissions are mitigated (Selivanova et al., 2024; Peng et al., 2020). Nevertheless, this decline is impacting the Arctic sea ice differently depending on regions and seasons, overall inducing a gradual shift from consolidated to seasonal sea ice conditions (Rolph et al., 2020). The present work focuses on changes in the Arctic marginal ice zone (MIZ), the transition region from the open ocean to the consolidated sea ice, traditionally defined as the region of the sea ice cover influenced by ocean waves (Horvat et al., 2020). Multiple definitions of MIZ have been advanced in the past. For instance, Sutherland and Dumont (2018) proposed to outline the MIZ extend through

the relative strength of wind and waves, whereas Dumont et al. (2011) combined the wave-ice interaction with the floe size distribution, defining the maximum ice floe size in the MIZ smaller than 200 m. Here, we use the common definition of MIZ as the region covered by 15% to 80% ice concentration (Frew et al., 2023), able to provide a standardized measure for comparing observation products and simulation results in the Arctic (Rolph et al., 2020). The limitations of a threshold-based definition

of MIZ, when evaluated through passive microwave retrieval, are mainly connected with warm air intrusions (Rückert et al., 2023). Liquid water in the clouds, surface melting, and melt ponds feature a different microwave emissivity compared to sea ice and can temporarily decrease the retrieved sea ice concentration. However, the threshold-based definition of the MIZ is well suited for simulation based on continuous sea ice models used for producing the reanalyses, and has therefore been preferred in the present study. In addition, using constant thresholds facilitate the computations of the MIZ area fraction in a consistent

manner, enabling the evaluation of past, present, and future sea ice conditions (Horvat, 2021).

The physical processes relevant to the MIZ differ from those in the pack ice, defined as the region where the ice concentration exceeds 80%. In particular, the MIZ sea ice experiences strong dynamical interactions with ocean currents, storms, and storm-generated waves (Manucharyan and Thompson, 2017; Alberello et al., 2019; Kohout et al., 2014), which in turn can lead to rapid thermodynamic sea ice changes, including enhanced lateral melting in fragmented ice floes (Tsamados et al., 2015; Frew

et al., 2023). Due to such negatively impacting feedback mechanisms, the projected enhancement of MIZ extent in crucial months (Frew et al., 2023) may further accelerate the melting of Arctic sea ice with consequences on the climate system and the Arctic ecosystems. The MIZ is fundamental to support a variety of biogeochemical processes (Galí et al., 2021), and changes in its extension and seasonality imply modifications of atmospheric-ocean heat, mass, and gas exchanges, with the potential to affect the habitat of organisms that rely on partially ice-covered ocean conditions (Rolph et al., 2020).

Here, we study the seasonal and interannual variability of MIZ through the lens of global ocean and sea ice reanalyses, as well as remote sensing sea ice observations. Global ocean reanalyses (ORAs) supply consistent and comprehensive historical records of ocean and sea ice variables by informing ocean model simulations with in situ and satellite observations through data assimilation techniques. Employing data assimilation is beneficial for reanalyses since it constrains the model state and reduces biases related to shortcomings in the physical model formulation. This feature is particularly desirable for sea ice

variables, for which many studies have unveiled substantial deviations from expected observational ranges (Tsujino et al., 2020). Therefore, ORAs can be reliable datasets for monitoring the present and past states of the sea ice and ocean. Moreover, given their relatively extended time coverage, which can reach more than 40 years, ORAs are becoming essential for monitoring the long-term variations of climate indices in a global warming regime, especially in regions where ocean and ice observations are not uniform in time and space, as the polar Arctic Ocean. Despite the well-known benefit of using ORAs for ocean research

applications (Storto et al., 2019), their quality in reproducing sea ice has been tested in a limited number of studies, e.g. (Chevallier et al., 2017; Uotila et al., 2019), and their application in polar regions is mainly restricted to a few regional products (e.g., PIOMAS). In light of this, we argue that assessing the quality of global reanalyses at high latitudes is a needed and timely endeavor.

The aim of our study is dual. Firstly, we intend to prove the quality and usability of global ORAs in representing sea ice

in the Arctic region. In completing this task, we follow the footsteps of Chevallier et al. (2017), who firstly assessed the

representation of the Arctic sea ice in 14 global reanalyses by considering multiple variables (e.g., sea ice concentration, thickness, and velocity). This work found premature using the ensemble of global reanalyses for sea ice monitoring due to the large spread in sea ice and snow thicknesses among the ensemble members. This assumption was revised shortly after by Uotila et al. (2019), who proved the usefulness of the multi-model ensemble mean in studying the physical state of the sea ice and the polar marine environment. In particular, their work show that the ensemble mean computed from 10 ocean reanalyses (global and regional) typically has a deviation from observational estimates smaller than the anomaly of individual ensemble members. Secondly, we aim to use these reanalyses to improve our understanding of the MIZ and investigate its behaviour in the context of Arctic sea ice internal and forced variability. The time evolution of Arctic sea ice at seasonal and interannual scales will be explored through the Global ocean Reanalysis Ensemble Product (GREP version 2), supplied by the Copernicus Marine Service (CMS). GREP differs from other ORAs because of its ensemble approach, which could further reduce remaining model biases not mitigated by DA. This dataset allows investigating the potential benefits of a multi-system approach which could further reduce remaining model biases not mitigated by DA. GREP is compared against regional reanalyses and satellite observations, highlighting the differences in MIZ and pan-hemispheric metrics. GREP includes four global ocean and sea ice reanalyses at eddy-permitting resolution covering the period from 1993 to the present, and its ocean and sea ice state both at global and regional scales were validated in various studies (Masina et al., 2015; Storto et al., 2019; Iovino et al., 2022).

Specifically, the following research questions are discussed in our work:

1. Can the ensemble approach overcome the limitations of single reanalyses in representing Arctic sea ice variables?

2. How different are GREP ensemble members? Is there a seasonal dependence in ensemble spread? And how does this compare to discrepancies in observational datasets?

3. Is the reanalysis performance in representing the MIZ in line with that of pan-hemispheric metrics?

4. Can reanalyses help to better understand MIZ processes, also in light of the observation's shortcomings?

The layout of the paper is the following. Sec. 2 and Sec. 3 illustrate the ocean reanalyses and the observational datasets used in this work, detailing their main features. Sec. 4 presents results on total and marginal ice, displaying the spatial distribution of sea ice variables, seasonal cycles, interannual variability and long-term trends. In addition, the position of MIZ and the evolution of ice classes are evaluated on a long-term basis. Finally, Sec. 5 concludes the paper by framing the results in the context of the ongoing sea ice research.

## 2 Ocean reanalyses

### 2.1 Global products

GREP is composed of four global ocean and sea ice reanalyses: C-GLORSv7 (Storto et al., 2016), FOAM-GloSea5v13 (MacLachlan et al., 2014), GLORYS2v4 (Lellouche et al., 2013), and ORAS5 (Zuo et al., 2019). In this study, monthly means

of sea ice variables are used for individual reanalysis as well as the ensemble mean and spread, available through the CMS catalogue (product reference GLOBAL_MULTIYEAR_PHY_ENS_001_031). Tab. 1 summarizes the main characteristics of GREP ensemble members (global reanalyses) and regional reanalyses used in the present work. A detailed description of model setups and data assimilation methods can be found in the GREP Product User Manual (Desportes et al., 2022).

The four ocean-sea ice reanalyses members are all driven by the ECMWF ERA-Interim atmospheric reanalyses (Dee et al., 2011), are constrained by satellite and in situ observations, and assimilate the same variables for ocean and sea ice: sea surface temperature (SST), sea level anomalies (SLA), sea ice concentrations (SIC), and in situ temperature and salinity profiles T/S(z). GREP and its constituent reanalyses cover the altimetric period from 1993. As ocean component, they all use the NEMO model (Nucleus for European Modelling of the Ocean http://www.nemo-ocean.eu/), and adopt the global tri-polar ORCA025 grid at

eddy-permitting resolution, approximately 1/4°of horizontal resolution and 75 vertical levels. Although many physical and numerical schemes are similar in the four reanalyses, there are several significant changes including the ocean model version and some parameterizations, thus introducing differences in the four ocean model configurations. Three out of four reanalyses use LIM2 thermodynamic-dynamic sea ice model (Fichefet and Maqueda, 1997; Goosse and Fichefet, 1999) with a single ice thickness category; the remaining one (FOAM-GloSea5v13) uses CICE4.1 (Hunke and Lipscomb, 2010) with a higher

complexity of ice physics, e.g. the ice thickness distribution. Sea ice rheology is modelled with the Elastic–Viscous–Plastic (EVP) formulation (Hunke and Dukowicz, 1997) by all reanalyses' sea ice models except for LIM2 implemented within ORAS5.

Data assimilation methods of the ensemble members differ for the numerical scheme, frequency and assimilation time windows, input observational data sets, error definitions, and bias correction schemes (Iovino et al., 2022). This is true for both

ocean and sea ice assimilated variables, leading to enlarging the spread among ensemble members' products. Focusing on the sea ice variables, C-GLORSv7 and FOAM-GloSea5v13 share the same assimilated data set (OSI SAF, described later) but with different frequency windows. Differently, GLORYS2v4 and ORAS5 ingest SIC from IFREMER CERSAT (Ezraty et al., 2007) and CMS OSTIA (Good et al., 2020), respectively.

## 2.2   Regional products

Regional ocean reanalyses used in this work are the state-of-the-art PIOMAS and TOPAZ4b made available by the Polar Science Center and the Nansen Environmental and Remote Sensing Center (NERSC), respectively. As for GREP ensemble members, their main characteristics and references are summarized in Tab. 1. The two products differ in all examined features, from the ocean and sea ice models to the data assimilation method and the assimilated datasets. Here, we highlight that PIOMAS assimilates sea ice observations from NOAA/NSIDC Climate Data Record, a data set not ingested by any member of

the GREP ensemble, making PIOMAS a stricter term of comparison for GREP. Differently, TOPAZ4b shares the assimilated sea ice data set with C-GLORSv7 and FOAM-GloSea5v13, even though the adopted data assimilation methods are different.

---

[1]Reprocessed before 2008, analysis from 2008.

[2]Shift from HadISST to NSIDC in 1996.

[3]Assimilation of CryoSat-2 SMOS sea ice thickness from 2010 onwards.

**Table 1.** Specifications of global ocean and regional reanalyses.

| Name | Global Reanalyses – GREP Ensemble Members | | | | Regional Reanalyses | |
|---|---|---|---|---|---|---|
| | **C-GLORSv7** | **GLORYS2v4** | **ORAS5** | **FOAM-GloSea5v13** | **PIOMAS** | **TOPAZ4b** |
| *Institution* | CMCC | Mercator Océan | ECMWF | UK Met Office | Polar Science Center | NERSC |
| *Ocean-sea ice model* | NEMO3.6-LIM2 (EVP rheology) | NEMO3.1-LIM2 (EVP rheology) | NEMO3.4-LIM2 (VP rheology) | NEMO3.2-CICE4.1 (EVP rheology) | POIM (VP rheology) | HYCOM-sea ice (EVP rheology) |
| *Time period* | 1986–2020 | 1993–2020 | 1976–2020 | 1993–2020 | 1976–2020 | 1991–2020 |
| *Ocean data assimilation method* | 3DVAR (7 days) | SAM2 (SEEK) (7 days) | 3DVAR-FGAT (5 days) | 3DVAR (1 day) | Nudging method | Deterministic Ensemble Kalman Filter |
| *Sea ice data assimilation method* | Linear nudging | Refused order KF (SEEK) | 3DVAR-FGAT | 3DVAR | Weighted nudging method | Deterministic Ensemble Kalman Filter |
| *DA sea ice data* | OSI SAF OSI-450-a 25 km | IFREMER CERSAT SSM/I sea ice conc. 12.5 km | OSTIA[1] SST_GLO_SST_L4_REP_OBSERVATIONS_010_011 0.05° | OSI SAF OSI-450-a 25 km | NSIDC[2] Near-Real-Time SSM/I-SSMIS 25 km | OSI SAF[3] OSI-450-a 25 km |
| *Thickness categories* | 1 | 1 | 1 | 5 | 12 | 1 |
| *References* | Storto et al. (2016) | Lellouche et al. (2013) | Zuo et al. (2019) | MacLachlan et al. (2014) | Zhang and Rothrock (2003) | Bleck (2002) |

The spatial resolution of the two products is 12.5 × 12.5 km for TOPAZ4b and <4/5>° for PIOMAS, where < > denotes the domain average. TOPAZ4b excludes the Bering Sea from the computational domain, thus underestimating the Northern Hemisphere total sea ice compared to all other reanalyses and satellite observations. The investigation of the Arctic sea ice displayed in this work is performed consistently with the TOPAZ4b domain and excludes the sea ice out of the Bering Strait from all reanalyses (the cut in latitude is performed at 67.5°).

## 3 Observational datasets

Performances of GREP, PIOMAS, and TOPAZ4b in computing the SIC are evaluated against two SIC observational datasets: the NSIDC Climate Data Record (version 4), hereafter NSIDC (Meier et al., 2021), and the OSI SAF Climate Data Record and Interim Climate Data Record (release 3), hereafter OSI SAF, product OSI-450-a (OSI SAF, 2022; Lavergne et al., 2019). The two products share a 25×25 km grid and monthly frequency, and both cover the period spanned by the reanalyses. The SIC is retrieved from the SMM/I and SSMIS instruments within 1993–2008 and 2006–2020, respectively. Both Climate Data Records (CDRs) use weather filters based on atmospheric reanalyses to minimize atmospheric disturbances on the retrieval. Despite their similarities, the products differ in their retrieval algorithm (Kern et al., 2022), which leads to different sea ice states, especially in winter, as we will show in the results section of the paper. As the MIZ can be impacted, we highlight that the NSIDC product is the combination of two well-established algorithms: the NASA Team (NT) algorithm (Cavalieri et al., 1984) and the Bootstrap (BT) algorithm (Comiso, 1986). Our choice of using two observational products as reference datasets is motivated by wanting a comparison as fair as possible between observations and reanalyses that assimilate observations from different sources.

Moreover, this work uses a satellite-derived dataset on the Arctic sea ice thickness (SIT), namely the merged CryoSat-2 SMOS (Soil Moisture and Ocean Salinity) version 203 provided by the Alfred Wegener Institute (Ricker et al., 2017). The SIT dataset, available between October and April, has already been used for model validations (Henke et al., 2023) and recently as the assimilated product in multiple global ocean-sea ice models (Cipollone et al., 2023; Cheng et al., 2023). Weekly optimally interpolated SIT are generated by merging two products with complementary characteristics. Radar altimeters on the polar-orbiting CryoSat-2 (Laxon et al., 2013) are efficient in determining the SIT thicker than 0.5 m (Zygmuntowska et al., 2014) by relying on snow-depth knowledge (Warren et al., 1999) and hydrostatic equilibrium assumption (Ricker et al., 2014; Tilling et al., 2016). Differently, SIT lower than about 0.5 m is derived from a passive microwave radiometer (Huntemann et al., 2014) within the European Space Agency (ESA) SMOS mission by evaluating the satellite brightness temperature in the L-band microwave frequency (Kaleschke et al., 2010). Here, we extract monthly averages of the CryoSat-2 SMOS product, and we interpret the SIT as an absolute thickness estimate, as performed in Cheng et al. (2023).

## 4 Results

### 4.1 Sea ice area seasonal and interannual variability at the panArctic scale

We begin by assessing the quality of global ocean and regional sea ice products against satellite observations at the panArctic scale. Figures 1(a, b) show the averaged March and September observed SIC for NSIDC between 1993 and 2020. The dashed and solid lines indicate the boundaries of the MIZ, highlighting the upper and lower SIC thresholds, respectively. March and September are chosen since the Arctic sea ice area (SIA) reaches the extremes of its seasonal cycle, cf. Fig. 2(a). In March (maximum of the seasonal cycle), the sea ice cover is composed almost totally of pack ice (SIC > 80%), and the MIZ is narrow and located in proximity to the sea ice edge. The pack ice sector is strongly reduced in September (minimum of seasonal cycle) and confined to the Central Arctic. During the summer months, the area of the MIZ grows and reaches approximately 30% of the entire SIA by September. It is worth noting that only NSIDC maps are displayed and used as reference since the OSI SAF, also considered in our analysis, exhibits very similar SIC patterns. Minor differences between the two datasets are that OSI SAF shows a slightly larger winter SIC in the Barents Sea and modestly lower values in the central Arctic in summer (not shown). The latter feature leads to a wider MIZ in OSI SAF compared to NSIDC due to the northward movement of the upper SIC threshold.

The SIC differences between the reanalyses and the NSIDC are shown in Fig. 1: maps from (c) to (e) display March, and maps from (f) to (h) display September. In general, there is a good agreement between the NSIDC and all reanalyses in March. GREP slightly underestimates the NSIDC SIC in the Central Arctic, while TOPAZ and PIOMAS slightly overestimate it, with the latter being the closest reanalysis product to the observational reference. Notably, less accordance is shown close to the Arctic sea ice edge, particularly at lower latitudes in the Atlantic and Pacific sectors. We recall that TOPAZ reanalysis does not include the sea ice positioned out of the Bering Strait due to the construction of the model domain; therefore, differences against TOPAZ are not displayed in the Pacific sector in Fig. 1(d) and (g). In September, the most significant differences between the reanalyses and observations are exhibited in the Central Arctic. All the reanalyses report underestimations of

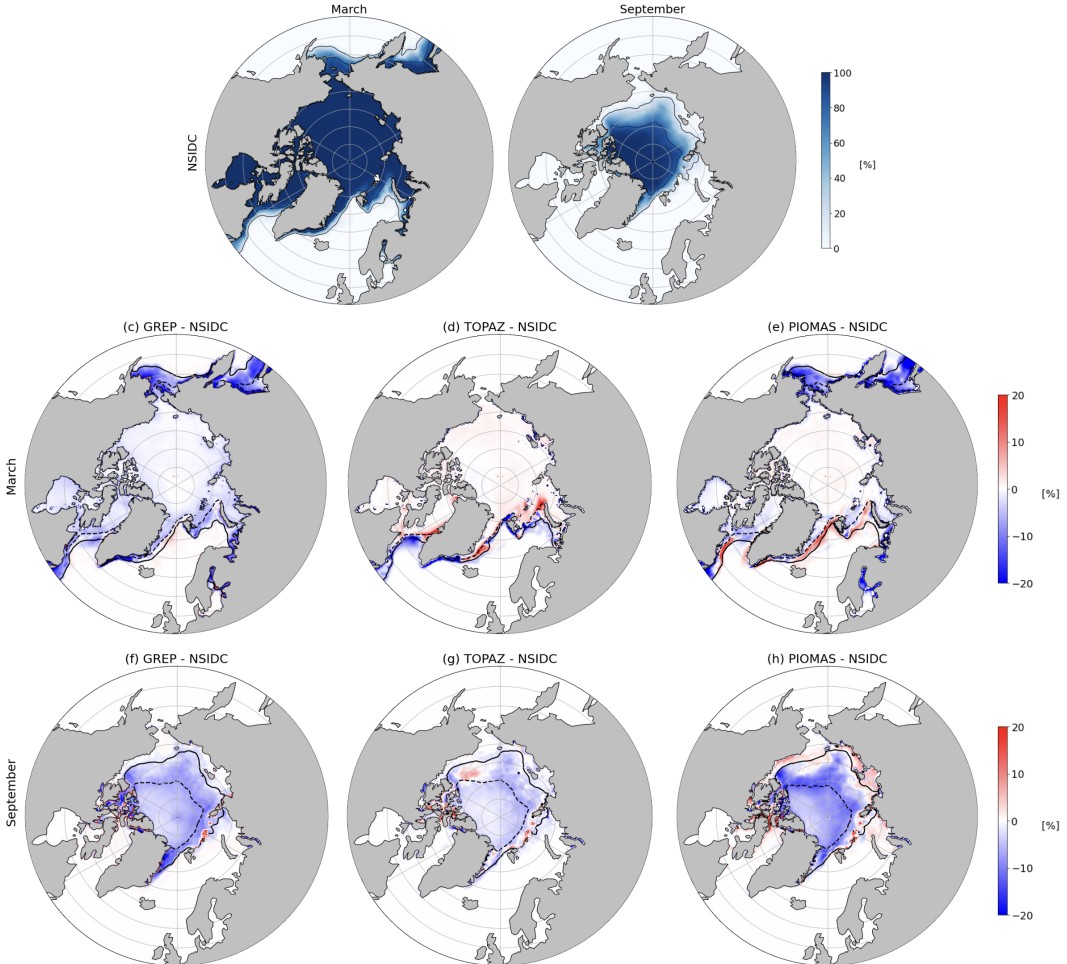

**Figure 1.** Monthly means of NSIDC SIC in March (a) and September (b) over the period 1993–2020. Differences in SIC between global/regional reanalyses and NSIDC in March (c-e) and September (f-h). GREP, TOPAZ, and PIOMAS are displayed from left to right. Differences in the sea ice outside the Bering Strait are shown if included within the reanalysis domain. In all panels, solid and dashed lines indicate respectively SIC = 15% and SIC = 80% evaluated from NSIDC (a-b) and ocean reanalyses (c-h).

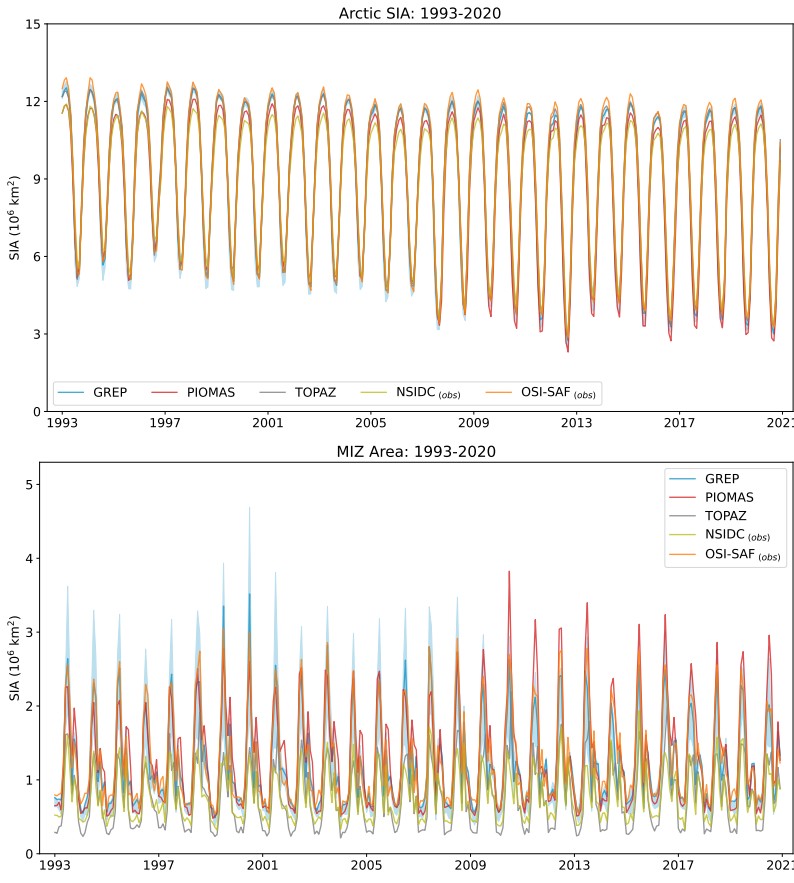

**Figure 2.** (a) Time series of monthly-averaged Arctic SIA and (b) Arctic MIZ over the period 1993-2020. Light blue shading depicts GREP members' envelope (the same in all akin figures).

observed SIC, with PIOMAS being the product that underestimates the SIC the most and TOPAZ the closest to the satellite reference. Few regions where reanalyses overestimate the observations are shown. They include areas between the Canadian
175 archipelago's islands for all products, a portion of the Beaufort Sea for TOPAZ, and the Siberian and Alaska shelves for PIOMAS. Importantly, these regions slightly overlap the MIZ, as shown by threshold contours.

Figure 2(a) shows the time series of the monthly-averaged total Arctic SIA in March and September for the global ocean and regional reanalyses and the satellite products. The SIA is computed as $\mathrm{SIA} = \sum_i \mathrm{SIC}_i A_i$, where the index $i$ runs over the cells of the domain and $A$ for the cell area. GREP ensemble mean (blue line) convincingly reproduces the interannual variability
180 of the SIA in these months: its SIA regularly falls within the range of the two satellite-based CDRs (orange and olive green lines) in March, and it frequently does so in September. It happens despite the spread of the GREP members (blue envelope) is typically broader than the range of satellite products in September. The opposite behaviour is exhibited in March when the spread of satellite-based CDRs is large enough to include all GREP members. Moreover, the GREP product captures the

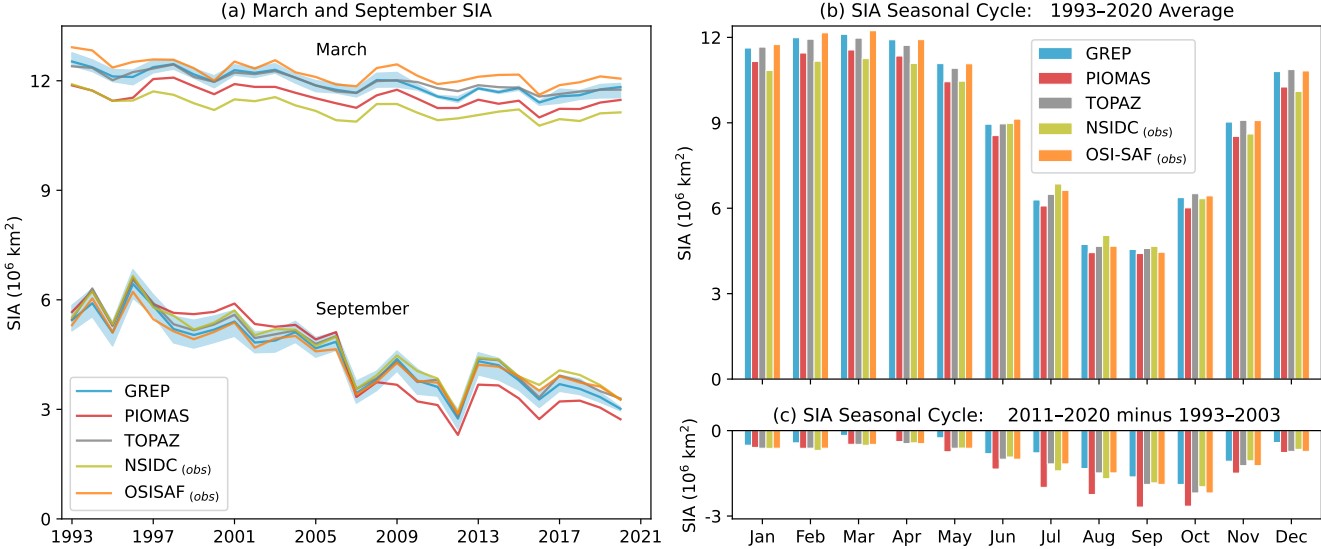

**Figure 3.** (a) Time series of monthly-averaged total Arctic SIA over the period 1993–2020 for March and September. (b) Seasonal cycle of total Arctic SIA computed for the same period and (c) differences between the last (2011–2020) and first (1993–2002) decades.

extreme events in summer, such as the strong minima in September 2007 and 2012. The two regional reanalyses (PIOMAS in red and TOPAZ in grey line) are also in good agreement with the observations during the winter maxima, falling consistently within the range of the two satellite products. This pattern also holds for TOPAZ at the end of the melting season, whilst PIOMAS shows a consistent underestimation pattern developing from 2009 onward. We will examine this behaviour in the concluding remarks.

The decreasing trend in SIA is quantified by the results in Tab. 2, showing the March and September total Arctic SIA trends computed within the period 1993–2020 for the global and regional reanalyses, and the satellite observations. The SIA trends for March and September indicate that the Arctic sea ice loss is larger in summer, which is in good agreement with what is shown, for example, in figure 1 in Matveeva and Semenov (2022) and table 1 in Wang et al. (2020), although the analyzed periods do not share the same initial and final dates. For GREP, March and September trends are respectively -0.31 $\pm$ 0.04 and -1.03 $\pm$ 0.11 $10^6$ km$^2$/decade, and they are very similar to what emerges from the satellite products. Also, the TOPAZ performance is acceptable in both months, while PIOMAS strongly overestimates the declining trend in September. The latter feature reflects the late summer SIA underestimation seen in Fig. 3a from 2010. The yearly-averaged SIA trends computed from the entire time series of monthly SIA over 1993–2020 (cf. Fig. 2(a)) complement the information displayed in the table. The trend in GREP is -0.68 $\pm$ 0.19 $10^6$ km$^2$/decade, which is in good agreement with those of observations and regional reanalyses, falling within one standard deviation from all of them. GREP result is boosted by the fact that Lee et al. (2023) uses the SIA trend -0.69 $10^6$ km$^2$/decade for the period 1997–2014 as a reference for assessing the panArctic accelerated rate of sea ice decline.

**Table 2.** Decadal trends of total SIA for the analyzed datasets within the period 1993–2020. Trends for March and September, and overall trends are shown.

| | March $10^6$ km$^2$/decade | September $10^6$ km$^2$/decade | Annual $10^6$ km$^2$/decade |
|---|---|---|---|
| GREP | -0.31 ± 0.04 | -1.03 ± 0.11 | -0.68 ± 0.19 |
| PIOMAS | -0.23 ± 0.05 | -1.42 ± 0.13 | -0.77 ± 0.18 |
| TOPAZ | -0.26 ± 0.03 | -1.03 ± 0.11 | -0.63 ± 0.19 |
| NSIDC | -0.28 ± 0.04 | -0.99 ± 0.11 | -0.62 ± 0.16 |
| OSI SAF | -0.28 ± 0.05 | -0.90 ± 0.10 | -0.61 ± 0.19 |

We conclude our total Arctic SIA-focused analysis by investigating the seasonal cycle. Fig. 3(b) shows the seasonality of the total SIA averaged over 1993–2020. The colors of the histogram bars, indicating different reanalyses or observational products, correspond to those in Fig. 3(a) and in the upcoming Fig. 3(c). Shared features are noticeable in all the products, with
confirmed minima and maxima SIA occurring in September and March and the same timing for growing and melting seasons. The seasonality of GREP compares well with TOPAZ and OSI SAF between November and May, while PIOMAS and NSIDC are closer to each other and show less extended SIA. This behaviour is reasonable since PIOMAS reanalysis assimilates the SIC from the NSIDC set of observations, while the other reanalyses assimilate OSI SAF data. In the remaining months, except for July and August, GREP is additionally close to the NSIDC, whereas PIOMAS SIA is still lower than other products. Fig. 3(c)
illustrates the differences between the seasonal cycle computed from the last (2011–2020) and the first (1993–2002) decade of the analyzed period. These differences indicate that approximately one-third of the initial summer SIA has been lost over the analyzed period, while lower portions are lost in the other months. For instance, in winter, the sea ice experiences a decrease of significantly less than 10% in terms of SIA. This visualization facilitates the previously observed underestimation pattern in PIOMAS SIA during the summer after 2010, as monthly differences are visibly larger than those of other products between
June and October.

## 4.2 The marginal ice zone at hemispheric scale

The ability of GREP to reproduce the MIZ is evaluated with the same approach adopted for the total Arctic SIA. The time series of monthly Arctic MIZ area from 1993 to 2020 is shown in Fig. 2(b). The MIZ area shows a clear seasonal cycle, with consistent winter minima lower than 1 $10^6$ km$^2$ and summer maxima that can reach up to half of the total SIA, cf. top panel of
the same figure. A second smaller peak is visible in October (the beginning of the freezing season), likely due to the response of the thin and not consolidated sea ice cover to Arctic cyclones (Serreze, 2009; Hutter et al., 2019; Blanchard-Wrigglesworth et al., 2022). The simulated GREP MIZ area falls almost always within the observed range of the two reference CDRs (NSIDC and OSI SAF), suggesting that GREP can provide a robust MIZ representation. Interestingly, this happens throughout the entire period, although the spread of GREP members at annual maxima is constantly wider than the observation products'
range before 2009.

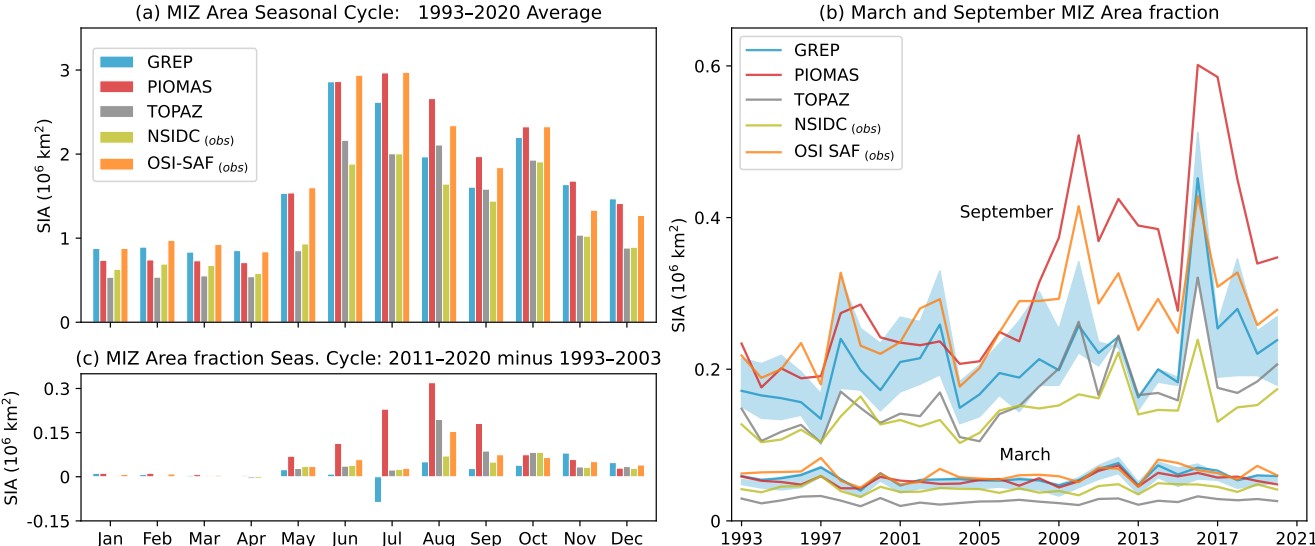

**Figure 4.** (a) Seasonal cycle of MIZ SIA computed within 1993–2020. (b) Time series of monthly MIZ area fraction over the same period for March and September. (c) Differences in MIZ area fraction between the last (2011–2020) and first (1993–2002) decade.

In winter, the observational range encapsulates all the reanalyses except for TOPAZ, which exhibits a systematic underestimation compared to GREP and PIOMAS. The minima occur between January and April, when the SIA is at its maximum and pack ice is bounded by the coastlines of the Arctic Ocean. Hence, the winter MIZ is representative of the low SIC conditions in the Barents Sea, the Kara Sea, the Atlantic Arctic sector, the Greenland Sea, and finally in the Baffin Bay and Labrador Sea
(Fig. 1a). It is worth noting that MIZ is similarly present also in the Pacific Ocean, but we omit its description because outside of our study domain. Maxima of MIZ area occurring between June and July exhibit a much higher degree of year-to-year variability. In these months, the reanalyses and CDRs are not fully compatible. From 1993 to 2009, the GREP ensemble mean occasionally shows the largest MIZ area, a behaviour driven by the extended differences between the four ensemble members. After 2009, the agreement between GREP and the CDRs improves, whereas PIOMAS performance substantially degrades and
exhibits MIZ area overestimation.

Figure 4(a) illustrates the seasonal cycle of the MIZ area within the period 1993–2020 for global and regional reanalyses and observations. The seasonality of the MIZ area aligns with the previously conducted analysis, with the GREP ensemble mean generally matching with PIOMAS and OSI SAF, except from July to September when GREP shows better agreement with TOPAZ and NSIDC. The latter rank close to each other within the entire seasonal cycle except in August.

Given the present context of the Arctic regime transition, the analysis of MIZ properties and variability is essential in light of the Arctic sea ice decline. Therefore, in Fig. 4(b,c), we introduce the MIZ area fraction as the percentage of the Arctic SIA formed of marginal ice. The September MIZ area shows a clear positive interannual trend for all the products, cf. Fig. 4(b). It contrasts the declining trend of total SIA; however, it is intuitively explained by the increasing summer tem-

**Table 3.** Decadal trends of monthly-averaged latitude of MIZ displayed in Fig. 5 (period 1993–2020) for GREP.

| | Jan | Mar | Jul | Sep |
|---|---|---|---|---|
| °/decade | $1.00 \pm 0.28$ | $0.69 \pm 0.24$ | $0.70 \pm 0.20$ | $1.42 \pm 0.25$ |

peratures, the enhanced sea ice melting, and the growing fragmentation of summer sea ice floes over most of the Arctic. Interestingly, the positive trend of the MIZ area fraction contrasts also with the trend of MIZ SIA in September, which results in $-0.053 \pm 0.046$ $10^6$ km²/decade for GREP. Combining the information, one can conclude that despite the area of MIZ shrinking with that of the total sea ice, its significance within the sea ice at the hemispheric scale increases. The behaviour is not valid throughout the year: in March, no noticeable tendency is observed for both MIZ area fraction (Fig. 4(b)) and MIZ area, which trend results in $0.014 \pm 0.023$ $10^6$ km²/decade. The finding confirms results in Rolph et al. (2020).

Figure 4(c) shows the difference in the seasonal cycle of MIZ area fraction between the last (2011–2020) and first (1993–2001) decades of the analyzed period. All products show almost no differences from January to April, corresponding to the period of the year with minimum values of MIZ area. From May to December, positive differences in the MIZ area fraction are evident in all datasets, except for GREP in July. Hence, for these months, the portion of sea ice falling within the MIZ class increased in recent years. When comparing the products, PIOMAS clearly overestimates this metric, whereas GREP generally displays moderate differences between decades, in line with NSIDC and OSI SAF.

The study of MIZ is complemented by the computation of its monthly-averaged latitude, a metric useful for quantifying the changes in the position of the marginal ice. Fig. 5 illustrates the evolution of MIZ monthly mean latitude for representative months from 1993 to 2020, with panels (a,b) and (c,d) showing winter and summer months, respectively. Before computing the average latitude, products are interpolated on the GREP grid. This makes the calculation resolution-independent, avoiding that grid points' distribution at lower latitudes affects the results producing spread between products. Clear positive trends are observed in all selected months and products, describing the northward movements of the MIZ during the last decades. Trends computed for GREP reanalysis are displayed in Tab. 3. We observed that they are smaller when the sea ice is at its maximum seasonal extension, e.g. March (Fig. 5(b)) and February (not shown), being the latitude of MIZ constrained by the presence of the Arctic coastlines. The trends are slightly higher when we consider the beginning of winter and summer, i.e. January in Fig. 5(a) and July in Fig. 5(c). Interestingly, the trend for July (characterized by an extended MIZ and pronounced sea ice melting) is very close to that observed in March, although showing noticeably different interannual MIZ latitude mean, 74.2° against 68.5°. The largest latitude shift is seen when the SIA is at its annual minimum: the monthly trend is $1.42 \pm 0.25$ °/decade in September, cf. Fig. 5(d). Similar results are seen in August (not shown).

The GREP envelope is narrower in January and March compared to July and September. The similar behaviour observed in the spread among products suggests that the presence of coasts addresses the location of MIZ in winter, which is less prone to the variations among products visible in summer. In addition, these variations increase throughout the analyzed period; in September, for example, the spread among all products varies from about 0.8° to 2.7°. This difference is driven by PIOMAS,

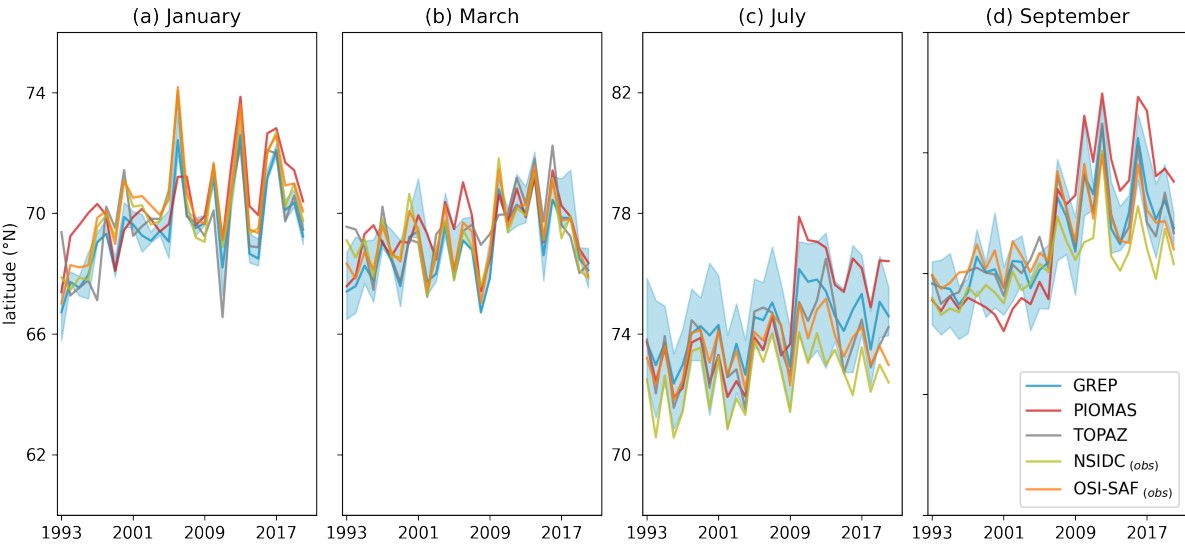

**Figure 5.** Time series of monthly-averaged latitudes of MIZ in winter (January, February, March), upper panels, and in summer (July, August and September), lower panels. It is worth noting that the Y-axis changes from (a,b) to (c,d).

that clearly displays larger values of MIZ average latitude after 2010 compared to all other products in summer, although generally preserving the time series pattern.

Overall, sufficient agreement among the products is shown, with differences among products not easily interpretable. In July, for example, NSIDC underestimates the latitude compared to all other products during the entire period of analysis, whereas in September it happens not continuously since PIOMAS displays a southernmost MIZ average latitude between 1997 and 2006.

### 4.3   Sea ice thickness for Arctic total and marginal ice

Figure 6 shows the monthly SIT averaged in March and September over the period 1993–2020. As one may expect, the thickest
sea ice is found north of Greenland and north of the Canadian Archipelago for all the reanalyses products. The products display slightly larger differences in other locations. In March, GREP exhibits thicker sea ice in the Beaufort Sea, PIOMAS in the Chukchi Sea, while TOPAZ shows lower SIT than the other products. In September, GREP SIT is greater than 2.5 m just in proximity to the Northern islands of the Canadian Archipelago. PIOMAS reproduces a region of thick ice that extends further towards the North Pole and slightly underestimates the SIT in the East Siberian Sea. Overall, GREP is in closer agreement with
PIOMAS in March and with TOPAZ in September. This happens despite two out of four GREP members assimilating the OSI SAF dataset, as TOPAZ does.

The time evolution of SIT for the total sea ice is displayed in Fig. 7 for the three reanalyses and satellite estimates based on the merged CryoSat-2 SMOS product. The satellite product, which covers only from 2011 onward, fits well the periodic annual pattern shown by all reanalyses (defined by November minima and May maxima) and presents the negatively-trended

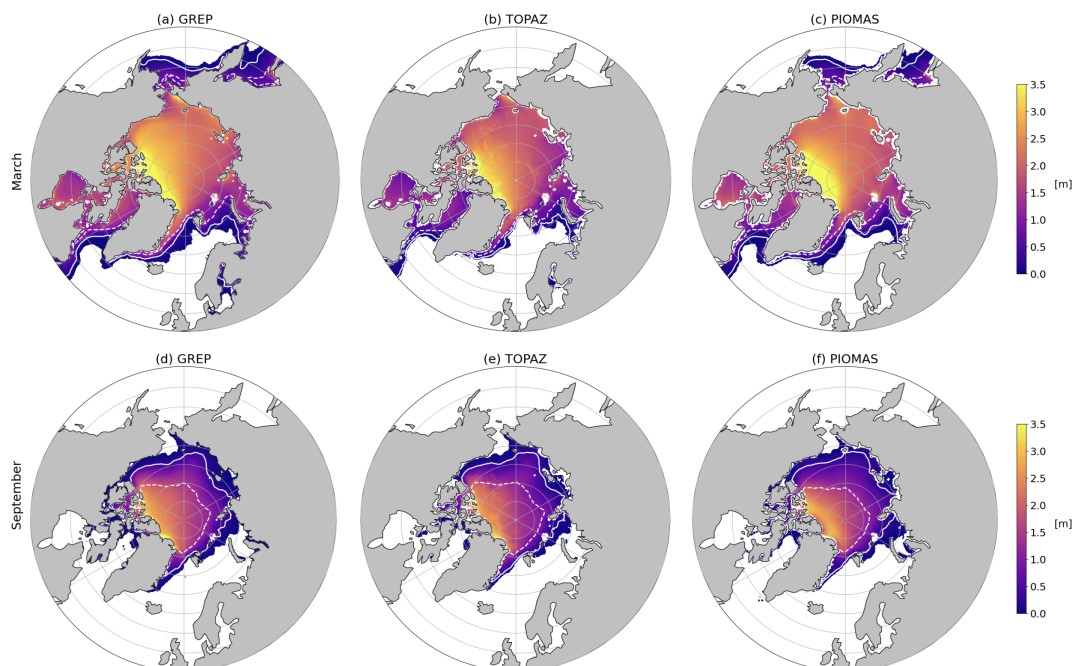

**Figure 6.** Monthly mean of SIT for GREP, PIOMAS, and TOPAZ in March (a-c) and September (d-f) over the period 1993–2020. Solid and dashed white lines display the thresholds of the MIZ, SIC = 15% and SIC = 80% respectively. Sea ice outside the Bering Strait is shown if available.

**Table 4.** Decadal trends of total SIT for global ocean and regional reanalyses within the period 1993–2020.

|  | m/decade |
| --- | --- |
| GREP | -0.26 $\pm$ 0.02 |
| PIOMAS | -0.30 $\pm$ 0.02 |
| TOPAZ | -0.07 $\pm$ 0.02 |

interannual variability shown by GREP and PIOMAS. These trends, computed over 1993–2020, show fair agreement with each other, with a weak negative trend from TOPAZ thickness (Tab. 4). However, trends estimated from GREP and PIOMAS can be considered more robust since TOPAZ substantially underestimates SIT from 1993 to approximately 2007. After 2007, the agreement with other reanalyses improves, even though TOPAZ still underestimates the SIT. The envelope of GREP members changes throughout the analyzed period: it widens from 1997 to 2002 and slowly narrows until 2011. PIOMAS is almost
constantly included in GREP shading and is very close to the GREP ensemble mean.

    To understand the link between SIA and SIT at the hemispheric scale, Figure 8 shows scatter plots of the monthly SIT versus SIA (left plots) and distributions of seasonal SIA according to SIT (right plots) for GREP (a,b), PIOMAS (c,d), and TOPAZ (e,f). The markers in the scatter plots indicate the monthly averages computed between 1993 and 2020, with different symbols and colors referring to various months and seasons respectively. The interannual variability is quantified via multi-year averages

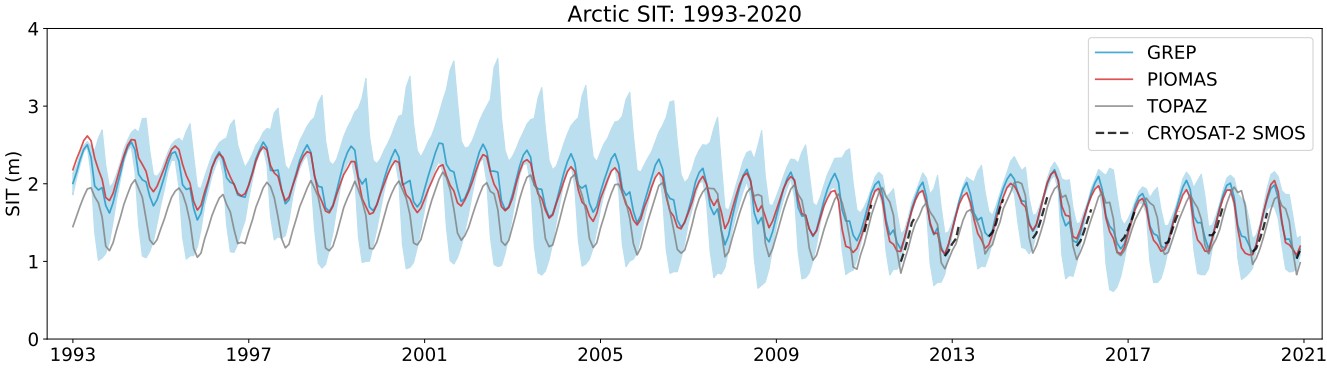

**Figure 7.** Monthly-averaged SIT for the total Arctic sea ice over the period 1993–2020. GREP, PIOMAS, and TOPAZ reanalyses are depicted in solid lines, GREP envelope in shading, and CryoSat-2 SMOS merged dataset in black dashed line.

over the analyzed period (depicted as highlighted symbols) and by the associated standard deviations. GREP (a) and PIOMAS (c) show a simultaneous increases in SIT and SIA from late summer (September) to winter (March). The mean SIT increases slowly until December, because of thin ice formation in open ocean, and afterward more rapidly in winter (blue cluster), when sea ice thickens in ice covered regions. In contrast, the SIA increases rapidly initially but is limited by the coastlines later in winter. TOPAZ (e) shows the same behavior, except that the multi-year average depicts a decrease in SIT from September to October. From April to the summer, all products show a decrease in both SIT and SIA, with thinning of sea ice and stable SIA during April and May, and almost stable conditions in August and September. In this period, TOPAZ overestimates the mean SIT compared to GREP and PIOMAS, as highlighted by the detachment of yellow and pink clusters from the green one. At the same time, TOPAZ shows a more compacted cluster on the x-axis direction, indicating an underestimation of the SIA annual cycle. From Fig. 8 GREP appears thinner than TOPAZ and PIOMAS in most seasons. This is not consistent with what displayed in Fig. 7, where GREP is well aligned with PIOMAS while TOPAZ underestimates the sea ice thickness. This discrepancy is motivated by the fact that the thickness of Fig. 8 is simply averaged, while in Fig. 7 we perform a weighted average based on the local sea ice concentration.

The right column of Fig. 8 displays the seasonal distribution of the local SIA as a function of its thickness. The plot is created by quantifying the SIA that falls within discrete thickness intervals of 20 cm. Therefore, the area below the curves is not normalized (thus different for each curve) and corresponds by construction to the total SIA averaged over a specific season. GREP and PIOMAS are in accordance regarding the shape of all seasonal distributions: winter and spring have well-defined peaks between 1.2 m and 2.2 m and smooth decline for thicker sea ice. Summer and autumn have less prominent peaks accompanied by local maxima for sea ice thinner than 20 cm (summer) and between 1.60 m and 2 m (autumn). As for the scatter plots, TOPAZ shows different features than the other reanalyses. The winter distribution exhibits a less prominent peak shifted toward thinner SIT ($\approx 80$ cm) as oppose to the summer distribution, with a larger peak moderately shifted toward larger SIT values. The spring distribution of TOPAZ is closer to those in GREP and PIOMAS, whereas the autumn curve behaves differently, mainly for sea ice thinner than 50 cm.

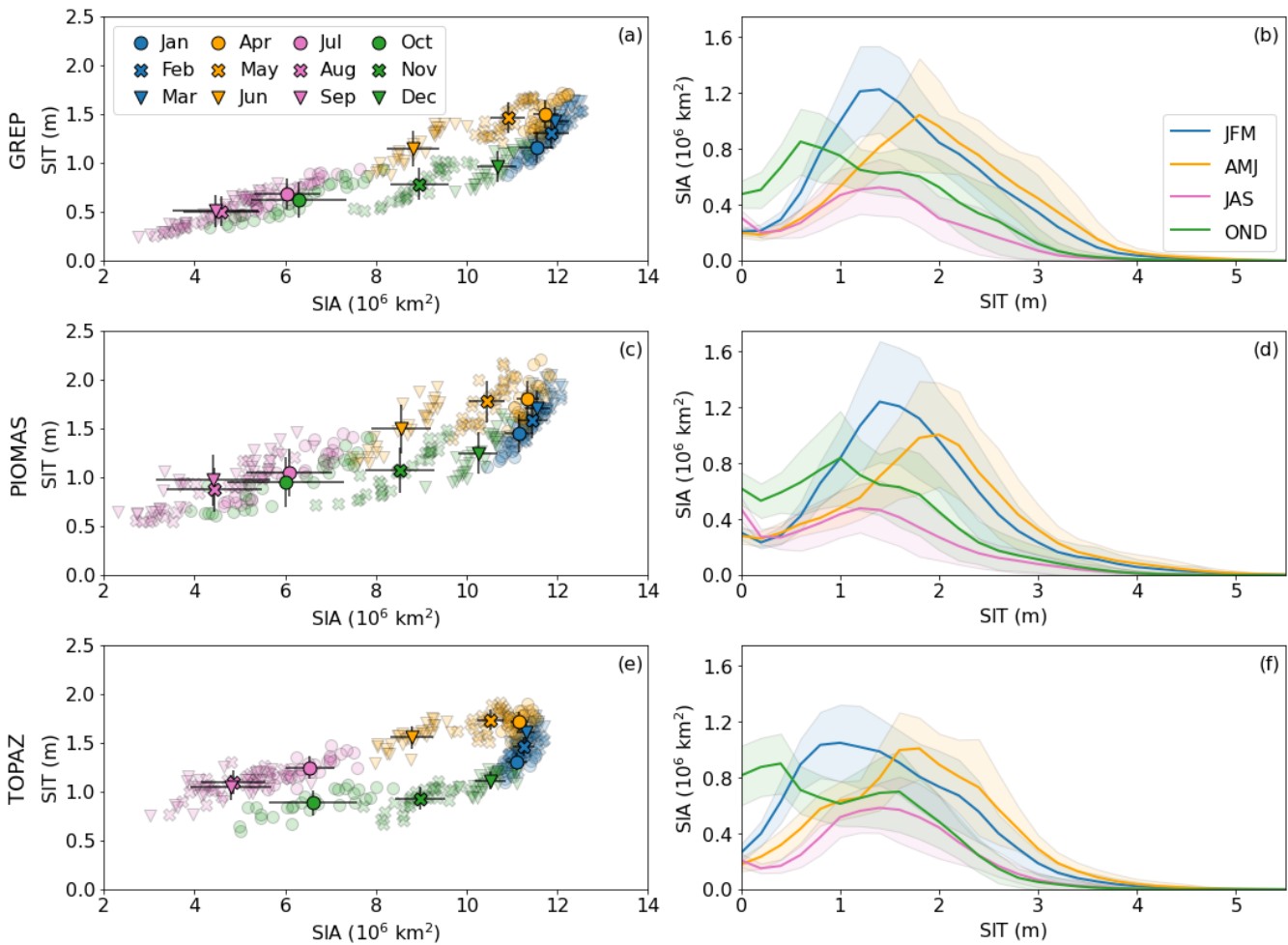

**Figure 8.** Left panels: scatter plots of monthly SIT versus SIA for GREP (a), PIOMAS (c), and TOPAZ (e) datasets. Scatter points in the background illustrate monthly averages between 1993 and 2020, colors and symbols allow to distinguish seasons and months. Highlighted symbols denote multi-year averages and bars the associated standard deviations. Right panels: seasonal cumulative distribution of SIA as function of SIT for GREP (b), PIOMAS (d), and TOPAZ (f) datasets; bold lines display the seasonal means while shadings indicate the standard deviations. The distribution was calculated by computing the SIA that falls within discreet thickness intervals of 20 cm.

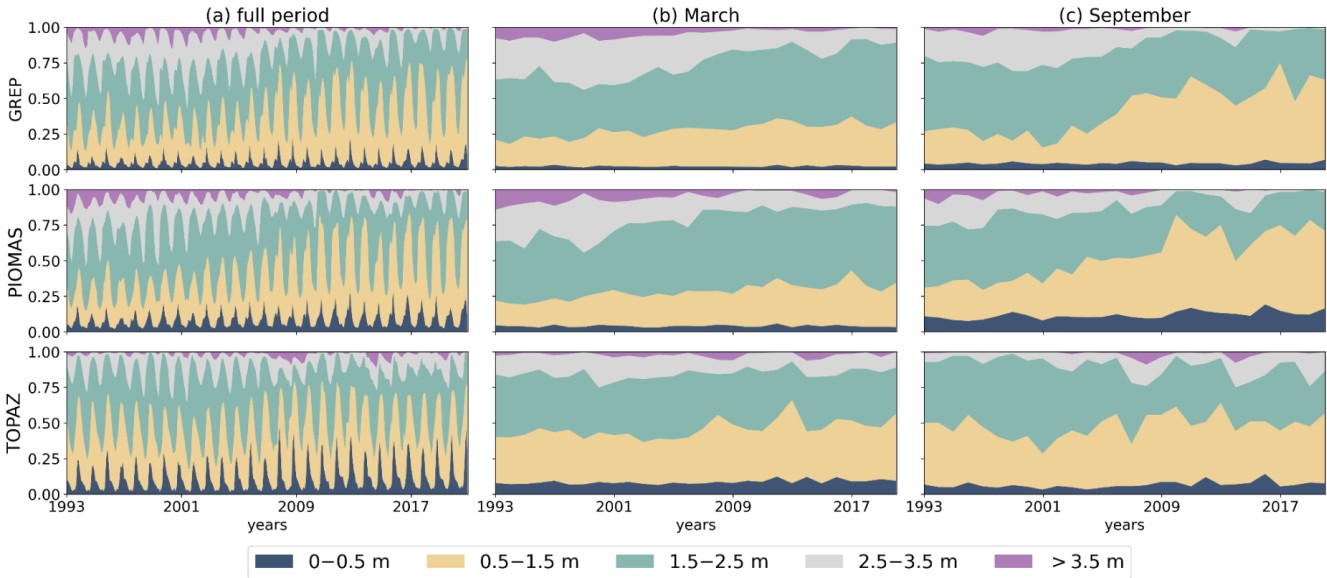

**Figure 9.** Evolution of the ice thickness categories shown by means of the fraction of total SIA for GREP, PIOMAS, and TOPAZ reanalyses. Panels (a), (b), and (c) display the analysis for the full period, March, and September.

Figure 9 shows the evolution of arbitrarily chosen thickness ranges evaluated as fractions of the total SIA. Rows organize the results according to the reanalysis, and columns differentiate the results achieved for the entire period (first column) and

selected months (March and September in the second and third columns, respectively). In the first column, the distribution of ice categories within the Arctic sea ice follows a seasonal cycle characterized by the largest presence of thicker sea ice categories in March and April and a prevalence of thinner sea ice categories in autumn.

These plots illustrate the changes in the presence of sea ice thickness categories during the analyzed period, comparing GREP against the regional reanalyses. In GREP, the presence of the thinnest ice category within the Arctic sea ice (dark blue

area) is reasonably stable across the time spanned by the analysis. Instead, distinct and moderate increases in the autumn peaks are visible in TOPAZ starting from 2005 and PIOMAS after 2010, respectively. GREP and PIOMAS indicate that the seasonal cycle amplitude for the ice category 0.5-1.5 m (yellow area) experiences a significant increase starting from 2002 as a result of the autumn category growth at the expense of thicker ice categories. In GREP, the fraction of sea ice covered by the ice category 0.5-1.5 m increases in GREP from about 18% to 35% in March and from 25% to 60% in September. The ice category 1.5-2.5 m

(teal area) exhibits the opposite behaviour, with a decrease of the area fraction in autumn and an increment in early spring at the expense of thicker ice categories (the latter indicates the thinning of sea ice at its maximum extension). In GREP, the ice category 1.5-2.5 m increases from about 40% to 55% in March and decreases from 55% to 35% in September; differently, it shrinks in both March and September in TOPAZ. The two thickest ice categories (sea ice with thickness greater than 2.5 m, grey and purple areas) share consistent shrinking patterns and seasonal cycle amplitude contractions for GREP and PIOMAS.

According to GREP, the fraction of sea ice covered by the ice category 2.5-3.5 m in March goes from about 25% in 1993 to

10% in 2020, while, in September, this category reduces from 10% in 1993 to nearly disappear in the last years of the analyzed period (less than 5% in almost all the years from 2009 to present). Similarly, the sea ice with a thickness greater than 3.5 m decreases; however, the shrinking terminates with the almost permanent vanishing of the summer sea ice from 2009 on, and a drastic reduction in other seasons to the extent that it regularly disappears in recent years. Differently, TOPAZ exhibits only partially the described features, being these categories more stable across the decades; for example, the sea ice thicker than 3.5 m covers less than 5% already at the beginning of the analyzed period.

## 5    Concluding remarks

The work accomplished the two objectives anticipated in the introduction. First, it proved the accuracy of GREP in reproducing the multi-year evolution and the annual pattern of the Arctic sea ice and its MIZ component. GREP product is compared against available regional reanalyses and satellite observations, displaying overall agreement. Strong correspondence is found when evaluating the total Arctic sea ice, with the spreading of satellite products larger than GREP members' envelope in winter and opposite behaviour observed in summer. In both cases, the GREP ensemble mean is fairly close to the reference products. Evaluating the marginal ice zone, significant discrepancies among products (including GREP) are shown in summer, when this component is at its annual maximum. The pronounced differences between satellite products and reanalyses (noticeable despite data assimilation) highlight the present-day issues in representing the MIZ, claiming for software and remote sensing improvements to better depict its ongoing increasing trend and its response to external forces. Nevertheless, the good performance of GREP in evaluating the area of total and marginal ice proves the robustness of GREP and determines its suitability as a boundary and initial conditions in forecasting systems. Moreover, GREP demonstrates its consistency during the full length of the experiment. This is not trivial to achieve, for example, we displayed that PIOMAS underestimates the SIA after 2008. This behaviour is driven by the transition of the assimilated data from NSIDC to the Near Real Time product described in https://nsidc.org/data/nise/versions/5. Interestingly, Fig. 3(a) also displays that a precedent shift from HadISST1 to NSIDC in 1996 (Schweiger et al., 2011) led to overestimate the SIA until approximately 2006. GREP also provides a fair analysis of the SIT compared to regional reanalyses in terms of multi-year evolution, correlation with the SIA, and trends of ice thickness categories. We also displayed that GREP provides SIT in line with those of PIOMAS during the full analyzed period. Differently, the TOPAZ trend for the sea ice thickness is not robust since it is prone to underestimations while assimilating CryoSat-2 SMOS data from 2010 onwards.

Despite the pronounced differences in depicting the marginal ice, the work emphasizes the common seasonal and inter-annual patterns of this sea ice type. The increasing trend of MIZ outlined in our analysis is prominent only in summer months, as the proximity of sea ice to the coastlines limits its inter-annual variability in winter. We can ascribe the modest winter variability to two main reasons. First, the atmospheric temperature gradient tends to be sharp along the ice edge, causing open-ocean patches and leading to refreezing rapidly into pack ice. Second, the heat content changes at the surface ocean, particularly in the Atlantic sector, is lower than that of the central Arctic, dampening in turn the variability of the MIZ. Nevertheless, a

substantial level of regional variations not captured by our panArctic metric still occurs, primarily due to variations of the North Atlantic Oscillation (NAO) phase.

Together with the displayed objectives, the main lesson learned from this work is that the unique ensemble approach of GREP can overcome the issues of single reanalyses in computing sea ice quantities. This is particularly evident when considering the SIT, for which the spread of GREP members largely changes during the analyzed period but the ensemble mean is always close to that of PIOMAS and CryoSat-2 SMOS when available. Conversely, when the dispersion among GREP members is narrower than the spreading of observational datasets, such as in winter, the ensemble mean product enhances its reliability.

This behaviour can also be observed when considering only Arctic marginal ice, for which GREP provides a more robust estimation than a single observational dataset. Demonstrating the reliability of GREP sea ice variables is especially important when describing the highly changeable marginal ice, likely the predominant condition of the future Arctic sea ice.

Finally, the study performed in this paper becomes particularly relevant when considering that reanalyses are becoming key products for training innovative machine learning models for predictions and possibly climate applications. While this

transformation is for now mostly confined to the atmospheric field, it is proving extremely successful and we believe there is the potential for this approach to spread to other earth system components, including the sea ice (Eayrs et al., 2024). For this reason, there is a growing need for studies assessing the quality of the current generation of sea ice reanalyses so that they can be used with confidence and possibly improved in the upcoming years (Zampieri et al., 2023). We believe this paper is an important step towards this goal.

*Data availability.*

The ocean reanalyses and satellite observations used in this study are available online. GREP is available for download at the CMS web page https://doi.org/10.48670/moi-00024. *Data from GREP members have been individually downloaded from the institutions' websites.* Products of SIC and SIT from regional reanalyses used here have been downloaded from distributors' websites: http://psc.apl.uw.edu/research/projects/arctic-sea-ice-volume-anomaly/data/model_grid for PIOMAS and ftp://my.

cmems-du.eu/Core/ARCTIC_REANALYSIS_PHYS_002_003/ for TOPAZ. Links to CDRs are: https://nsidc.org/data/g02202/versions/4 for NSIDC and ftp://osisaf.met.no/reprocessed/ice/conc/v3p0/ for OSI SAF. Merged SIT dataset CryoSat-2 SMOS is downloaded from ftp://ftp.awi.de/sea_ice/product/cryosat2_smos/v203/nh//. CMS OSTIA dataset ingested by ORAS5 is the product SST_GLO_SST_L4_NRT_OBSERVATIONS_010_001 available at https://doi.org/10.48670/moi-00165. IFREMER CERSAT datset of SIC is available at ftp://ftp.ifremer.fr/ifremer/cersat/products/gridded/psi-concentration/. All scripts used

for the analysis are available at the repository (Cocetta, 2024).

*Author contributions.*

FC and LZ wrote the manuscript. FC and JS analyzed simulated and observed datasets. DI conceived and designed this study. All authors contributed to the interpretation of results, and edited and reviewed the manuscript.

*Competing interests.*

The contact author has declared that none of the authors has any competing interests.

*Acknowledgements.* We gratefully acknowledge Christian Haas and the two anonymous reviewers for their suggestions on how to improve the manuscript. We thank the present and past members of the CLIVAR/CliC Northern Oceans Region Panel for their pivotal role in enhancing our knowledge of the Arctic's influence on the Earth's climate.

*Financial support.* This research was supported by the European Union Horizon 2020 research and innovation programme under grant
agreement no. 101003826 via the project CRiceS (Climate Relevant interactions and feedbacks: the key role of sea ice and Snow in the polar and global climate system) and by the European Union via the ObsSea4Clim (Ocean observations and indicators for climate and assessments) project (grant Agreement number 101136548).

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
