# Peer review of "Assessing the Representation of Arctic Sea Ice and Marginal Ice Zone in Ocean/Sea Ice Reanalyses"

_EGUsphere, 2024_

## Author Comment (AC1)

Dear Editor and Reviewer,

We thank you for accurately reading and commenting on the manuscript and suggesting how to improve it. Detailed answers to each of your comments are provided hereafter. We hope you find them satisfactory. Reviewer comments are in black, followed by our response in blue, which includes changes and/or additions to the text.

For the authors,

Francesco Cocetta

The authors extend the evaluation of sea ice in the CMEMS GREP Ensemble Reanalysis Product from the Antarctic (Iovino et al., 2022) to the Arctic in this manuscript, focusing on panArctic scale performances and the Marginal Ice Zone (MIZ) properties. As MIZ is an increasing proportion of the Arctic sea ice regime under climate change, accurately representing their spatiotemporal variabilities is becoming a key benchmark for sea ice modeling skills.

Overall, this manuscript is well-structured and well-written. More importantly, the proposed scientific questions are sound and adequately discussed. I like the authors' indication of ocean-sea ice reanalysis application scenarios, especially for the hottest machine learning techniques at the moment, where reanalyzed data is not only complementary to observed data, but also may be a more recommended dataset for training models. This is another valuable guideline beyond the data quality assessment for reanalysis users.

I recommend the publication of this excellent manuscript after addressing the following detailed issues, which are rather minor:

- Introduction: The authors have thoroughly reviewed the Arctic sea ice changes under climate change and ORAs' role in this subject. However, I am unfortunate to find Chevallier et al. (2017) and Uotila et al. (2019), two very comprehensive assessments of sea ice in global/regional reanalyses, surprisingly uncited. I believe the Introduction will be completed by connecting your work with theirs.

Thank you for the suggestion which enabled us to enhance the introduction by linking our work more closely with state-of-the-art literature. In the revised version of the paper, we updated the text in lines 46 and lines 50-56, accordingly.

- Line 129: "(SMOS)" firstly appears in line 122.

Thank you. We moved the explanation of the acronym from line 129 to line 122.

- Line 135: "NSIDC" has been defined in line 110.

Agreed, thank you. We removed the explanation of the acronym in line 135.

- Line 144: "wide" should be "wider".

Done, thank you.

- Line 147: "(f) to (e)" should be "(f) to (h)".

Agreed, thank you.

- Line 158: "Figure 3(a)" should be "Figure 2(a)"?

Agreed, thank you.

- Line 160: The seasonal cycles in Fig. 2(a) contain information on both climatology and interannual variability, with the former dominating the curve. This makes it difficult to signal interannual variability directly from the figure. I wonder if drawing anomalies would be more intuitive.

We acknowledge this limitation of the plot. However, plotting the time series of the sea ice area as it is, helps us to illustrate when annual minima and maxima occur, justifying the maps of March and September we show in Fig. 1.

Following the other reviewer's comment, in the new version of the paper, we will make the plots in Fig.2 easier to read by shortening the x-axis while extending the y-axis as follows. Below is also the plot of SIA anomalies.

[Figure]

[Figure]

- Figure 3: Although you have mentioned the meaning of the individual line types in the text, I suggest that it would be clearer to the reader if you also specify it in the figure legend.

Implemented, thank you for the comment.

- Line 163-164: It seems to me that September has the smallest spread, within the range of two observations, while March has the largest.

Yes, the spread of the GREP envelope is larger in September compared to March, when it fits within the range of the two observations, which is larger in March.

- Line 200: Does "1 $10_6$ km2" mean "1x$10_6$ km2"?

Yes, we leave this to the typesetting phase.

- Line 198: "Fig. 2(b)" should be "Fig. 4(b)".

Fig. 2(b) is correct. However, we acknowledge that "variations" can be misleading if referred to that figure. Therefore, we use "time series".

- Line 204-205: I find it difficult to understand this sentence, please rephrase it.

Thank you for the comment. Here, we wanted to highlight that, for almost the entire period, the GREP ensemble mean is within the range of observed products at annual maxima, although the spread of GREP members is constantly larger than the observed products' range before 2009.

We rephrased the sentence as follows: "Interestingly, this happens throughout the entire period, although the spread of GREP members at annual maxima is constantly wider than the observation products' range before 2009".

- Figure 4: It might be better to rearrange the subfigures in Fig. 4 in the order in which they were written in the main text.

Considering that Fig. 2(b) in line 198 is correct, the arrangement of subfigures in Fig. 4 is OK.

- Table 3: Why not directly list these four quantities in Fig. 5?

Thank you for the comment. We tried to follow the suggestion but we found it difficult since we would have to move the legend to the top of the July panel and the result is not appealing.

- Line 225: I do not get what "MIZ SIA" means.

Thank you for the comment. We replaced the wording with MIZ area.

- Line 255: Should the section number read "4.3"?

Agreed. Thank you.

- Table 4: Also, I would recommend directly listing these three quantities in Fig. 7.

We keep the table detached from the figure in consistency with the choice made for Table 3.

- Additionally, I recommend labeling the SICs throughout the text in %, including the colorbars in figures.

Thank you for the suggestion. In the revised version of the paper, we switched from fraction numbers to percentages when describing SIC.

References

Chevallier, M., Smith, G. C., Dupont, F., Lemieux, J.-F., Forget, G., Fujii, Y., et al. (2017). Intercomparison of the Arctic sea ice cover in global ocean–sea ice reanalyses from the ORA-IP project. *Climate Dynamics*, *49*(3), 1107–1136. https://doi.org/10.1007/s00382-016-2985-y

Uotila, P., Goosse, H., Haines, K., Chevallier, M., Barthélemy, A., Bricaud, C., et al. (2019). An assessment of ten ocean reanalyses in the polar regions. *Climate Dynamics*, *52*(3–4), 1613–1650. https://doi.org/10.1007/s00382-018-4242-z

---

## Author Comment (AC2)

Dear Editor and Reviewer,

We thank you for accurately reading and commenting on the manuscript and suggesting how to improve it. Detailed answers to each of your comments are provided hereafter. We hope you find them satisfactory. Reviewer comments are in black, followed by our response in blue, which includes changes and/or additions to the text.

For the authors,

Francesco Cocetta

The study by Cocetta et al. shows by comparison with regional reanalysis products (PIOMAS and TOPAZ) as well as some other satellite products (CS2/SMOS or SIC from OSI SAF and NSIDC) that the global Ocean Reanalysis Ensemble Product (GREP) is very well able to resolve regional and interannual variability and trends in ice cover and ice thickness. In particular, a comparison of the ice thicknesses of different reference products with GREP emphasizes the additional value of using ensemble means.

To my knowledge, the comparison of global ensemble means with regional reanalysis data is new and innovative. The good agreement is worth mentioning as reanalysis products are becoming increasingly important for a range of scientific application (e.g. training of models). The paper is well written, very clearly structured and has a number of interesting figures. This paper shows that existing sea ice reanalysis products can be trusted, at least with respect to the variables studied here.

In parts, the authors could try to better explain observed differences or refer to other studies if necessary. I find the paper very interesting and recommend it for publication if the following points are addressed:

I find the definition for MIZ difficult. As described by the authors, the MIZ is also characterised by its proximity to the open ocean. I am afraid that the threshold-based classification (15-80%), even if based on a 25 km product, is not sufficiently accurate and induces errors. I recommend that the authors either:

- A) to show that the definition used is sufficiently accurate and enables a MIZ classification even in years when the ice concentration in parts of the central Arctic falls below 80%.

- B) to expand/revise the definition of the MIZ and, for example, to take into account the distance to the ice edge

- C) To refrain from using the term MIZ and instead speak of "low ice concentration areas" or similar.

Thank you for the comment. We acknowledge that the definition of the marginal ice zone (MIZ) used in the analysis is just one of many formulations available for quantifying this

unique zone. Over the past decades, several criteria have emerged for characterizing the MIZ, yet finding a coherent definition remains a challenge. The MIZ was originally described as the region where polar air, ice, and water masses interact with the ocean temperature and subpolar climate system (Wadhams et al. 1981). More recently, it has been commonly defined as the portion of the ice-covered ocean where surface gravity waves significantly impact the dynamics of sea ice (e.g. Wadhams 2000), typically featuring highly variable ice conditions.  For instance, Dumont et al. (2011), considering the influence of ocean waves in breaking up sea ice into smaller floes, defined the MIZ as the region where the maximum ice floe size is less than 200m. Sutherland and Dumont (2018) proposed a definition of the MIZ extent based on the relative strength of wind and waves. Due to the unknowns in wave-ice interaction and the large uncertainties in both observed and simulated wave propagation within sea ice, the MIZ in model studies can be operationally characterized using sea ice concentration thresholds and defined as the transition zone between open water and consolidated pack ice, where the total area of the ocean is covered by $15\% - 80\%$ sea ice (e.g. Paul et al. 2021). This criterion, commonly used in model studies  (references provided in the paper), provides a standardized measure for comparing observation products and model sea ice results in the Arctic region (Rolph, 2020).

We acknowledge the limitations of using this MIZ definition for the Arctic observations, particularly in winter when warm air intrusions may cause a SIC decrease (Ruckert et al., 2023). However, these errors generally affect passive microwave retrieval for up to a few days (weather timescales) and do not significantly impact the monthly sea ice concentrations used in our study. Moreover, Ruckert et al. (2023) studied a warm air intrusion in April 2020 and showed that NSIDC and OSI SAF climate data records (assimilated by reanalyses investigated in this paper and directly used here for comparisons) are less prone to SIC decrease compared to NASA team and ASI products.

The adopted definition of MIZ is less restrictive in summer, when SIC retrieval algorithms may be affected by surface melting, melt ponds, and liquid water present in clouds. In these circumstances, the sea ice concentration may be underestimated, leading the MIZ extent to include portions of fragmented sea ice that are not affected by waves and/or are not close to the open ocean.

Bearing in mind these limitations, we decided to perform our analysis on the Arctic MIZ using the threshold-based definition, which is directly applicable to the output of model simulations. Furthermore, our analysis includes the use of the MIZ area fraction, which has been proven to be suitable for evaluating past, present, and projected sea ice conditions (Horvat, 2021).

To address the legitimate concern of the reviewer, we expanded the first paragraph of the introduction clarifying the shortcomings associated with our definition of MIZ and arguing its utility despite these limitations. We hope that this will help the readers of the paper to contextualize our results.

This answer is intended to address also some of the detailed comments below, namely the comments to lines 25 and 204, and to chapter 4.1.

I also wonder whether the title of the manuscript should not be kept more general. The focus in the title on the MIZ is not really necessary, as a much broader comparison is made that goes well beyond the MIZ

Thank you for the idea; we realized that the focus of our analyses is broader than the solely marginal ice zone. Therefore, we decided to reformulate the title into "Assessing the Representation of Arctic Sea Ice and Marginal Ice Zone in Ocean/Sea Ice Reanalyses". In our view, the updated title better reflects the content of the paper, by mentioning both the total sea ice and the marginal ice zone and shifting the emphasis from GREP to ocean/ice reanalyses.

More detailed comments:

Abstract is well-written and presents the main objectives and findings.

Line 4: GREP: The Climate Copernicus website usually refers to the Global ocean Reanalysis Ensemble Product. I would leave out "ocean".

The GREP version, available via Copernicus Marine Service and used in this study, is described in the following user manual https://catalogue.marine.copernicus.eu/documents/PUM/CMEMS-GLO-PUM-001-026.pdf.

The complete name has changed over time on the Copernicus webpage and on the available documentation. For example, the GREP acronym can be found extended as Global Ocean Ensemble Physics Reanalysis, Global Ocean Ensemble Reanalysis product, and Global Reanalysis Multi-Model Ensemble Product. We apologize for any potential inconsistency. In the manuscript, we used a definition widely accepted in the global reanalysis community from which the acronym originally came. To address the reviewer's comment, we updated the product reference, which effectively defines the data set in the CMS catalog, to the new version GLOBAL_MULTIYEAR_PHY_ENS_001_031.

Line 1: May be use "data-assimilating" instead

Suggestion accepted. Thank you.

Line 8: Widening or expanding? Or both?

Thank you for the comment. Following it, in the revised version of the paper we slightly changed this sentence referring directly to the increase of the percentage of marginal ice zone (i.e. MIZ fraction) in summer.

Line 15: Transition

Done. Thank you.

Line 16: Provide more recent reference

Thank you for the suggestion. We added three more recent references: Shokr M. and Ye Y. (2023) for sea ice extent, Sumata et al. (2023) for sea ice thickness, and Babb et al. (2023) for multi-year ice. Complete references are displayed in the bibliography of the revised version of the paper.

Line 20: Suggestion: "…are expected to continue unless anthropogenic greenhouse gas emissions are mitigated"

We have implemented the suggestion, thank you.

Line 25: This definition of MIZ seems somewhat outdated to me (see general comment above). In the summer months in particular, ice concentration values well below 80 % can occur temporarily throughout the Arctic. This is the result of intensified melting but sometimes also atmospheric effects (e.g. https://doi.org/10.1525/elementa.2023.00039 ). Is there a way to take this into account in the definition? And if not, what impact is this likely to have on the validation?

This point has been addressed while answering the first general comment.

Line 30: "projected future expansion"…reference missing! Rolph primarily take a look at the satellite era. Won't the MIZ area automatically become smaller at some point the less ice remains in the Arctic in the summer months? For example, if ice is limited to the last ice areas in a few years' time, wouldn't the MIZ be much smaller than it is today?

We rephrased the sentence and added the reference as follows:

Due to such negatively impacting feedback mechanisms, the projected enhancement of MIZ extent in crucial months (Frew et al., 2023) may further accelerate the melting of Arctic sea ice with consequences on the climate system and the Arctic ecosystems.

Line 38: DA = Data Assimilation?

Yes, thank you. We refer to Data Assimilation. We corrected the text in the reviewed paper.

Line 31: Dissimilar: Consider using "different"

Done, thank you.

Comment: The research questions at the end are nicely shaped

Line 74: I was searching the CMS catalogue but could not find any product via the product reference.

We updated the product reference to the renamed version GLOBAL_MULTIYEAR_PHY_ENS_001_031.

Line 76: … but is there a reference that describes model setup and data assimilation method? Can you add it?

Thank you for the comment. For all reanalyses included in the GREP product, the model setup and data assimilation method are described in the references displayed in Table 1. Namely, Storto et al. (2015) for C-GLORSv7, Lellouche et al. (2013) for GLORYS2v4, Zuo et al. (2019) for ORAS5, and MacLachlan et al. (2014) for FOAM-GloSea5v13.

Additionally, the Product User Manual (PUM) of GREP (Desportes et al., 2022) provides an overview of all reanalyses by listing the members' models, data assimilation (DA) schemes and assimilated variables along with output frequency and common features. We will also include the reference to the PUM in the revised paper.

Tab. 1 and text: DA sea ice data: Please provide product reference and coverage/resolution for remote sensing observations. E.g. OSI-450a / CERSAT, etc…

Thank you for the comment. We extended the information in the row "DA sea ice data" in Table1.

Tab 1: The provided reference refers to a documentation of the model?

Yes, the references provided are publications detailing the reanalysis setup and performances. For GREP members, except ORAS5, the references listed in the table derive from the GREP Product User Manual. For ORAS5, we cited a more recent paper by the same author.

Chapter 4.1: I find the definition of the MIZ somewhat difficult. As the author points out in the intro, the MIZ is characterized by wave/tidal interaction and the influence of the open ocean. I believe that a simple threshold value (i.e. ice concentration of less than 80%) is not a sufficiently precise definition, but that other parameters should actually be included here. This could be, for example, the distance to the open ocean or from the ice edge. See my general comment above. I think a good summary of the MIZ definition problem is given in https://journals.ametsoc.org/view/journals/atot/34/7/jtech-d-16-0171.1.xml

See our answer to the first general comment.

General comment: If I have understood correctly, all the products used have the same resolution (1/4°, or 25 km). Consequently, no major differences in the product comparisons are to be expected that could be attributed to the resolution? Perhaps it would be good to emphasise this again.

Thank you for the comment. The global GREP members all have a horizontal resolution of 1/4°, which is different from that of regional reanalyses: TOPAZ resolution is 12.5 X 12.5 km and PIOMAS average resolution is 4/5°.

In the revised version of the paper, we added the resolution of regional reanalyses in section 2.2.

Fig. 2b): It is somewhat difficult to see the differences. Maybe you show any anomalies here? Alternatively you can shorten the x-axis and make the y-axis larger.

Thank you for the suggestion. Accordingly, we modified the size of the plot as reported below.

[Figure]

Line 148: I could imagine that the reanalysis products together with the applied MIZ definition provide a better estimate of the MIZ area than the satellite data.

Fig 3: Very nice performance. Good and interesting to see

Line 173: Fig. – figure, Tab. – table

Thank you for the suggestion. When referring to figures and tables in cited papers, we prefer to use the full words to mark the difference.

Line 189: I think family does not fit.

Agreed, we switch to "set" in the revised version.

Line 201: interesting observation! Just out of curiosity: I wonder if the frequency or magnitude of such autumn peaks has increased over time?

From Fig.2(b) there is no clear trend in magnitude or frequency in the autumn peaks of MIZ.

Line 204: Again, I am not sure if satellite observations can provide a robust MIZ estimate with the applied definition.

This point has been considered while answering the first general comment.

Line 221: … add "in Fig. 4 c)

Thank you. Modification done.

Line 222: I agree, although these are processes that affect the entire Arctic and not just the MIZ.

Line 225 and following: I believe that an interpretation of the trends is difficult due to the MIZ definition used.

Thank you for the comment. In this paragraph, we compare the trend of the MIZ area with the trend of the MIZ area fraction in September. We point out that while the former is negative, indicating a narrowing of the MIZ area, the latter is positive, highlighting the increasing presence (and importance) of the MIZ relative to the total sea ice.

Fig. 5: I may have missed it: but I would expect a smaller spread in January and March MIZ latitudes compared to July and September averages. Can you explain why this is? Is it because the area of the MIZ is smaller in March and limited to areas characterized by high variability (Fram Strait)?

Thank you for the comment. Yes, in January and March, the MIZ (in its annual minimum, cf. Fig. 4(a)) is located Southernmost compared to July and September in areas with high variability such as Fram Strait but also Barents Sea, Davis and Denmark Straits, cf. Fig 1(a). However, there could be the possibility that the larger spread in January and March depends on the distribution of the grid points at lower latitudes. Therefore, we are working on interpolating the products on the GREP grid and drawing the plot again. If there are differences compared to the present version of the figure, we will update it providing a thorough description.

Fig. 7: Is it possible to enlarge Y-axis?

Thank you for the suggestion. We updated the figure with more readable axes.

Line 270: …it slowly narrows until 2011: Any idea why this is?

The narrowing is given by the improved performance of GREP members, in particular by GLORYS and ORAS5 which are responsible for the wide envelope in the central years of simulations. In these years, GLORYS provides overestimated values of SIT and ORAS5 underestimates the minima in SIT.

Line 275-285: Interesting scatter plot! However, I think it would be important to go beyond a description of the agreements and initiate a discussion about the larger spread in the summer months, for example. This could be done with reference to other studies

that look at similar relationships (Fig. 3; lag correlation between sea ice volume and thickness: https://www.nature.com/articles/s41586-022-05058-5)

Thank you for the comment. We acknowledge the importance of initiating a discussion on the correlation between Sea Ice Thickness (SIT) and Sea Ice Area (SIA) in this context. Therefore, we decided to enhance and expand Figure 8 as shown below, with each row corresponding to a different reanalysis product.

[Figure]

*Figure 8. Left panels: scatter plots of monthly SIT versus SIA for GREP (a), PIOMAS (c), and TOPAZ (e) datasets. Scatter points in the background illustrate monthly averages between 1993 and 2020, colors and symbols allow to distinguish seasons and months. Highlighted symbols denote multi-year averages and bars the associated standard deviations. Right panels: seasonal distribution of SIA according to SIT for GREP (b), PIOMAS (d), and TOPAZ (f) datasets; bold lines display the seasonal means and shadings the standard deviations. Plots are drawn by computing the SIA within thickness intervals of 20 cm.*

In the left column panels, we present SIT versus the SIA as in the previous version, but with the y-axis depicting the mean SIT rather than the weighted-mean SIT. This modification allows for a clearer explanation of the sea ice thickening throughout the seasons. In addition, we included different symbols for each month (alongside colors for seasons) and plotted the multiyear monthly means with associated standard deviations for both SIT and SIA.

On the right, we display the distribution of SIA categorized by ice thickness. For each season, we plot the mean and the standard deviation. The plot was produced by

computing the SIA for thickness intervals of 20 cm, mimicking a distribution function plot; we used lines to facilitate the visualization and comparison between seasons.

Lines 275-285 in the revised paper describe the new figure.

Fig. 9 and discussion: That's a very interesting comparison! Nice to see. Perhaps you can adjust the colour display in Fig. 9. Red/green/orange is difficult to distinguish.

Thank you for the comment. We chose a different palette which allows better distinction between colors.